# *CpBBX19*, a B-Box Transcription Factor Gene of *Chimonanthus praecox,* Improves Salt and Drought Tolerance in *Arabidopsis*

**DOI:** 10.3390/genes12091456

**Published:** 2021-09-21

**Authors:** Huafeng Wu, Xia Wang, Yinzhu Cao, Haiyuan Zhang, Run Hua, Huamin Liu, Shunzhao Sui

**Affiliations:** 1Chongqing Engineering Research Center for Floriculture, Key Laboratory of Horticulture Science for Southern Mountainous Regions of Ministry of Education, College of Horticulture and Landscape Architecture, Southwest University, Chongqing 400715, China; whfwhf@email.swu.edu.cn (H.W.); wx221069@email.swu.edu.cn (X.W.); yinzhu202108@163.com (Y.C.); zhy469703581@163.com (H.Z.); naitang911007@163.com (R.H.); 2College of Landscape Architecture and Life Science/Institute of Special Plants, Chongqing University of Arts and Sciences, Yongchuan, Chongqing 402160, China; liuhuanin@126.com

**Keywords:** *CpBBX19*, *Chimonanthus praecox*, salt stress, drought, overexpression, *Arabidopsis*

## Abstract

Zinc-finger proteins are important transcription factors in plants, responding to adversity and regulating the growth and development of plants. However, the roles of the *BBX* gene family of zinc-finger proteins in wintersweet (*Chimonanthus praecox*) have yet to be elucidated. In this study, a group IV subfamily *BBX* gene, *CpBBX19*, was identified and isolated from wintersweet. Quantitative real-time PCR (qRT-PCR) analyses revealed that *CpBBX19* was expressed in all tissues and that expression was highest in cotyledons and inner petals. *CpBBX19* was also expressed in all flower development stages, with the highest expression detected in early initiating bloom, followed by late initiating bloom and bloom. In addition, the expression of *CpBBX19* was induced by different abiotic stress (cold, heat, NaCl, and drought) and hormone (ABA and MeJA) treatments. Heterologous expression of *CpBBX19* in *Arabidopsis thaliana* (*Arabidopsis*) enhanced the tolerance of this plant to salt and drought stress as electrolyte leakage and malondialdehyde (MDA) concentrations in transgenic *Arabidopsis* after stress treatments were significantly lower than those in wild-type (WT) plants. In conclusion, this research demonstrated that *CpBBX19* plays a role in the abiotic stress tolerance of wintersweet. These findings lay a foundation for future studies on the *BBX* gene family of wintersweet and enrich understanding of the molecular mechanism of stress resistance in wintersweet.

## 1. Introduction

Wintersweet (*Chimonanthus praecox*), belonging to the Calycanthaceae family, is native to China, has a distinct flowering period (from November to March), strong aroma, and yellow flowers [1,2], and is widely used for cut flowers and as a garden plant in China [3]. In addition, wintersweet is used for aromatic oil extraction, Chinese herbal medicine, and cosmetics [1,2,4]. The unusual flowering period of wintersweet shows that this species has a strong ability to tolerate low temperatures [5]. Some genes and transcription factors have been identified as related to biotic and abiotic stress in wintersweet [6]. For instance, heterologous expression of *CpSIZ1* in *Arabidopsis* enhanced the cold resistance of transgenic *Arabidopsis* (*Arabidopsis thaliana*) [5], while overexpression of the gene *CpLEA5* increased the ability of transgenic *Arabidopsis* to resist low temperature and tolerate salt and alkali [7]. In addition, the amino acid sequence of CpFATB has high sequence similarity with the choroidal palmitoyl acyl carrier protein thioesterase of other plants, and overexpression of *CpFATB* in tobacco enhanced the drought tolerance of transgenic tobacco (*Nicotiana tobaccum*) [8]. Furthermore, CpCOR413 is a membrane-stable protein related to the cold tolerance of wintersweet [9], while overexpression of the gene *CpTAF10* increased the germination rate of transgenic *Arabidopsis* seeds, and under salt stress, the taproot and lateral roots of transgenic plants showed certain growth advantages compared with WT plants [10].

In plants, transcription factors, as key elements of the expression network, participate in the regulation of expression of numerous genes and are instrumental in regulating plant growth and development and the response to the external environment, including biotic and abiotic stresses [10,11,12]. The *BBX* gene family, encoding zinc-finger proteins, is an important family of transcription factors in plants, with functions including response to environmental changes, and plant growth and development [13,14]. BBX proteins generally have one or two B-box domains, and some members also have VP domains and CCT domains [15]. In plants, the B-box domain contains approximately 40 amino acids and is highly conserved, but there are still some differences among different B-box motifs [16,17]. The B-box domain binds to Zn ions to stabilize its unique tertiary structure [18], which is involved in protein–protein interactions [13]. For example, BBX1 (CONSTANS), BBX4 (COL3), BBX21 (STH2), BBX22 (STH3), BBX24 (STO), and BBX25 (STH) all interact with COP1 (CONSTITUTIVE PHOTOMORPHOGENESIS 1) [16,19,20,21,22]. The CCT domain of BBX proteins comprises 42 or 43 conserved amino acids and mainly mediates transcription regulation and protein transport, but also has a function in positioning. The VP domain is adjacent to the CCT domain and mainly mediates the interaction with *COP1* [17,18,23]. In *Arabidopsis*, the BBX protein family is divided into five structural types according to the number of B-box domains and CCT domains: the first type (*BBX1*-*6*) has two tandem B-box domains, one VP domain, and one CCT domain; the second type (*BBX7-13*) has two B-box domains and one CCT domain, but lacks a VP domain; the third type (*BBX14*-*17*) contains one B-box domain and one CCT domain; the fourth type (*BBX18*-*25*) only contains two tandem B-box domains, and the fifth type (*BBX26*-*32*) has only one B-box domain [17,18,23].

Under abiotic stresses, such as drought, high and low temperatures, and high salt, multiple *BBX* gene family members were strongly induced [17,24,25]. In *Arabidopsis*, the thermostability of AtBBX18-RNAi plants was improved, while the thermostability of *AtBBX18* overexpression plants was decreased. In *AtBBX18* overexpression plants, heat stress response genes *DGD1*, *Hsfa2*, and *Hsp101* were downregulated, suggesting that the gene *BBX18* plays a negative regulatory role in the heat-stress response of *Arabidopsis* [26]. Under drought stress, multiple *MdBBX* genes were upregulated in roots and leaves of apple trees (*Malus domestica*) [27]. Overexpression of *MdBBX10* enhanced the resistance of transgenic *Arabidopsis* to drought, salt, and other abiotic stresses, which was closely related to the enhancement of active oxygen-scavenging ability [28]. High salt and Polyethylene glycol (PEG) treatment induced *SsBBX24* gene expression and protein accumulation in *Solanum sogarandinum* [29]. Exogenous hormones treatment, such as abscisic acid (ABA), gibberellin (GA), ethylene, cytokinin, and auxin, also affected transcriptional levels of *BBX* genes [30,31,32]. The expression level of *CmBBX24* was influenced by GA4/7, and suppressing the expression of *CmBBX24* reduced cold and drought tolerance of *chrysanthemum* [33]. Under treatment with GA, Salicylic acid (SA), and Methyl Jasmonate (MeJA) hormones, expression levels of the genes *OsBBX2*, *OsBBX7*, *OsBBX17*, *OsBBX19*, and *OsBBX24* were upregulated at different time points [31]. Treatment of *Arabidopsis* leaves with ABA resulted in significant upregulation of *BBX11, BBX13*, and *BBX22* gene expression levels, while *BBX2, BBX3, BBX16, BBX18,* and *BBX19* genes were downregulated by cyclic adenosine diphosphate ribose (cADPR), which is involved in ABA early signal transduction [34].

Under adversity stress, the permeability of plant cell membranes changed. The degree of plant cell membrane damage could be judged by measuring the extravasation of cell electrolytes [35]. Malondialdehyde (MDA) is an important product of membrane lipid peroxidation and is usually used as an indicator of membrane lipid peroxidation. Therefore, the changes in electrolyte leakage and MDA content reflect the degree of damage to the membrane system under adversity conditions, and were widely used in the research of plant cold resistance, drought resistance, and salt–alkali tolerance [36,37].

The BBX protein family was widely reported to be involved in the process of abiotic stress in plants [10,11,12]. The functions of BBX zinc-finger proteins have been clarified in many plants, such as Arabidopsis [26], chrysanthemum [38], and apple [28]. However, the BBX gene family has not been extensively studied in woody plants such as wintersweet. According to information from the wintersweet transcriptome database [39], expression of a new *BBX* gene, *CpBBX19*, might be different under abiotic stress. In this study, we identified and isolated a *BBX* gene, *CpBBX19*, in wintersweet, and explored the expression patterns and function of this gene. *CpBBX19* was classified into the IV subfamily of BBX proteins, and significantly improved the tolerance of transgenic *Arabidopsis* to salt and drought.

## 2. Materials and Methods

### 2.1. Plant Materials

To analyze the expression pattern of the gene *CpBBX19* in wintersweet, the roots, stems, cotyledons, and young and old leaves were collected from six-leaf wintersweet, and flowers during the full bloom period were collected from five-year-old plants. Using the wintersweet floral development stages classification method [40], flowers of germination, flower-bud, petal-display, early initiating bloom, later initiating bloom, blooming, and withering were collected, respectively. The tissues were rapidly frozen in liquid nitrogen after collection. Three biological repeats were obtained for each sample.

Wild-type *Arabidopsis* (ecotype Columbia−0) was used for plant transformation. After sterilizing in hypochlorite solution, *Arabidopsis* seeds were sown on solid Murashige and Skoog (MS) medium, vernalized at 4 °C for 3 days, and then grown in an environment of 22 ± 1 °C with a 16 h light/8 h dark photoperiod. After 2 weeks, the seedlings were transferred into a matrix of vermiculite and peat (1:1 ratio) and cultivated from MS medium under the same conditions as described above.

### 2.2. Gene Isolation

Total RNA was extracted from the wintersweet tissues according to the manufacturer’s instructions of the RNAprep pure plant total RNA extraction kit (Tiangen, Beijing, China). To ensure RNA quality and quantity, RNA extracts were visualized by 1% agarose gel electrophoresis, then a Nano-Drop ND-1000 spectrophotometer (Thermo Fisher Scientific, Wilmington, MA, USA) was used for quantification at optical densities of 260 and 280 nm. The cDNA samples were synthesized using a PrimeScript RT reagent kit with gDNA Eraser (TaKaRa, Otsu, Japan). Each 10 µL contained 2 µL 5 × gDNA Eraser Buffer, 1 µL gDNA Eraser, and 1μg total RNA. Then, PCR procedure was performed PCR at 42 °C for 2 min; first-strand cDNA was synthesized by reverse transcriptase in 20 μL reaction system containing 10 µL reaction product from the previous step, 1 µL PrimeScript RT Enzyme Mix I, 4 µL 5 × PrimeScript Buffer 2, and 4 µL RNase Free dH_2_O [5]. The process of PCR was as follows: 37 °C for 15 min, then 85 °C for 5 s, and end at 4 °C. Specific primers *CpBBX19*-F/R (Appendix A) were designed by Primer Premier 6.0 to clone the DNA sequence of *CpBBX19*. The PCR procedure comprised an initial preheating step at 95 °C for 5 min, followed by 27 cycles of denaturation at 95 °C for 30 s, annealing at 55 °C for 30 s, and extension at 72 °C for 45 s, with a final extension at 72 °C for 10 min. PCR amplification products were separated via electrophoresis on a 1% agarose gel, and the target DNA fragments were recovered using an Agarose Gel DNA extraction kit (Tiangen) according to the manufacturer’s instructions. The resulting fragments were cloned into the pMD19-T vector (TakaRa, Dalian, China) and sequenced by TsingKe Company (TsingKe, Chengdu, China).

*BBX19* upstream promoter sequence was cloned using genomic walking method via Universal Genome Walker^TM^ 2.0 User Manual kit (Clontech, Mountain View, CA, USA) by manufacturer’s description. Specific primers (SP1 and SP2) (Appendix A) were designed according to the ORF sequence of *CpBBX19*. The genomic DNA walking library enzyme digestion template was established by the following steps: first, digest with EcoRV blunt-end enzyme, then passivate the digested product, and then ligate with T4-DNA ligase, and connect the upstream and downstream genome walking adapters in a 16 °C water bath overnight. The PCR reaction system contained 2.5 μL 10 × buffer, 0.5 μL dNTPs, 0.5 μL primer APF1/2, 0.5 μL primer SP1/2, 0.5 μL Taq DNA polymerase, 1 μL DNA, and 19.5 μL ddH_2_O. The reaction procedure was carried out according to the Manufacturer’s instructions. RNA was detected by 1% gel electrophoresis, the target band was recovered, and it was sequenced to determine the upstream sequence of *CpBBX19*.

### 2.3. Bioinformatics Analysis

The sequence of *CpBBX19* was determined using the BLAST-Protein program in the National Center for Biotechnology Information (NCBI) database, available online https://blast.ncbi.nlm.nih.gov/Blast.cgi (accessed on 25 December 2019). Multiple sequence alignments were performed by BioEdit software. The online analysis website MEME (The MEME Suite 5.3.3) was used for protein motif analysis; motif setting value was six; available online https://meme-suite.org/meme/info/status?service=MEME&id=appMEME_5.4.11632023872135256190210 (accessed on 8 July 2021). To understand the distribution of *Cis*-acting elements upstream of the *CpBBX19* promoter sequence, the 2000-bp sequence upstream of *CpBBX19* [41] was analyzed by PLANTCARE, available online http://bioinformatics.psb.ugent.be/webtools/plantcare/html/ (accessed on 19 March 2021).

### 2.4. Plant Treatments

Expression profiles of the gene *CpBBX19* were determined under different abiotic and hormonal treatments using the six-leaf plant as the materials [42]. Drought and NaCl treatments involved the use of 50% PEG-6000 [43] and 300 mM NaCl [31] irrigation, respectively. Cold and heat stress comprised treatments at 4 °C and 42 °C [44], respectively. For exogenous hormone treatment, 50 µM ABA and 100 µM MeJA were sprayed [42], respectively. There were three biological replicates per treatment. Samples were collected at 0, 2, 6, 12, and 24 h after treatment and quickly frozen in liquid nitrogen.

### 2.5. Quantitative Real-Time PCR (qRT-PCR)

*CpBBX19* expression patterns were analyzed in different tissues and flowers development stages of wintersweet. RNA was extracted from the wintersweet samples according to the method described in 2.2. The qRT-PCR analyses were performed by Bio-RAD CFX96 (Bio-RAD CFX Manager Software Version 1.6). Specific primers (RT-*CpBBX19*-F/R) were used for PCR amplification, and the process of qRT-PCR was performed according to the method described by Liu et al. [45]. According to the manufacturer’s instructions, Ssofast EvaGreen Supermix (50 × 20 μL reactions) includes 2× reaction buffer with dNTPs, Sso7d-fusion polymerase, MgCl_2_, EvaGreen dye, and stabilizers. Each 10 μL reaction mixture contained 5 μL Ssofast EvaGreen Supermix (Bio-RAD, Hercules, CA, USA), 0.5 μL each gene-specific primer, and 3.5 μL nuclease-free water. The PCR program comprised preheating at 95 °C for 30 s, followed by 40 cycles of denaturation at 95 °C for 5 s, annealing at 60 °C for 5 s, and extension at 72 °C for 5 s. There were three biological replicates for each experiment and three technical replicates for each sample. *CpActin* genes were employed as the internal reference gene of wintersweet [42]. Specific primers required for amplification were designed by Primer Premier 6.0 software. The qRT-PCR product of *CpBBX19* was 150 bp, and the primers (RT-*CpBBX19*-F/R) are shown in Appendix A.

### 2.6. Construction of Expression Vectors 

*CpBBX19* was fused in-frame into the *pGWB551* vector with the CaMV35S promoter by Gateway recombination reactions [43] to generate the 35S::*CpBBX19* construct. Specific primers (*pGWB551*-*CpBBX19*-F/R) used in the process are shown in Appendix A, and the plasmid map of the 35S::*pGWB551* vector is shown in Appendix A. The resulting 35S::*CpBBX19* plasmid was confirmed by sequencing and then transformed into *Agrobacterium tumefaciens* (strain *GV3101* competent cells) via electroporation to further explore the function of *CpBBX19*.

### 2.7. Generation of Transgenic Arabidopsis and Stress Treatments

*Arabidopsis* transgenic lines (OE lines) were obtained by transferring the 35S::*CpBBX19* recombinant plasmid into wild-type *Arabidopsis* via the floral dip method [46]. T-0 seeds were sown on MS medium containing 25 µg/mL hygromycin for transgenic selection. The method of Khan et al. [47] was used to extract DNA from the leaves of transgenic and nontransgenic plants (WT), and PCR amplification was performed using the extracted DNA and *CpBBX19* specific primers (*CpBBX19*-F1/R1) (Appendix A). The PCR procedure comprised an initial preheating step at 95 °C for 5 min, followed by 27 cycles of denaturation at 95 °C for 30 s, annealing at 55 °C for 30 s, and extension at 72 °C for 40 s, with a final extension at 72 °C for 10 min. PCR products were detected by 1% gel electrophoresis to confirm the insertion of *CpBBX19* into transgenic plants. qRT-PCR was used to identify the expression level of *CpBBX19* in the transgenic lines, with the *AtActin* gene [43] used as the internal reference gene of *Arabidopsis*. The qRT-PCR product of *AtActin* was 200 bp. Three T3 transgenic lines (with high, medium, and low expression of *CpBBX19*, respectively) were selected for phenotypic investigation; wild-type *Arabidopsis* (WT) was used as the control.

To determine the salt tolerance of the transgenic lines, WT and transgenic *Arabidopsis* were irrigated every 2 days with 300 mM NaCl to ensure that the concentration of NaCl solution in the tray remained uniform. To test the drought tolerance of the transgenic lines, WT and transgenic *Arabidopsis* were irrigated with 30% PEG-6000. MDA content and electrolyte leakage of transgenic *Arabidopsis* were detected as indicators of stress tolerance. MDA content was determined as described by Heath et al. [48], while the method of electrolyte leakage was stated by Seliem et al. [49].

### 2.8. Data Analysis

IBM SPSS Statistics 23 software package was used for statistical analysis, and Duncan’s multiple range tests were employed to analyze the significance of differences. A *p*-value < 0.05 was recognized as statistically significant, and a *p*-value < 0.01 was classed as extremely significant.

## 3. Results

### 3.1. Isolation and Characterization of CpBBX19

Based on the wintersweet transcriptome sequence in the database [39], a 1536-bp full-length cDNA of *CpBBX19* (accession number MZ740468) was obtained. The 5′ untranslated region (5′-UTR) and a 3′ untranslated region (3′-UTR) of *CpBBX19* were 475 bp and 428 bp, respectively, and the open reading frame (ORF) of *CpBBX19* was 633 bp, encoding 210 amino acids. The predicted molecular weight (MW) of the resulting protein was 23.47 kDa, and the deduced isoelectric point (PI) and instability index were 5.91 and 56.50, respectively. Subcellular localization prediction analysis of the *CpBBX19* gene-encoded protein of wintersweet showed that the protein was predominantly distributed in the nucleus.

Multiple sequence alignment revealed that the CpBBX19 protein sequence was highly homologous to the BBX19 protein sequence of *Arabidopsis*, *Gossypium hirsutum*, *Hibiscus syriacus*, *Ziziphus jujuba*, and *Hevea brasiliensis* (Figure 1a), indicating that BBX19 was conserved in plants. The CpBBX19 protein contained two B-box domains, but lacked the CCT domain and VP domain at the C-terminus. A phylogenetic tree of CpBBX19 and *Arabidopsis* BBX proteins was reconstructed by MEGE 6 (Figure 1b) and showed the five subfamilies of the BBX protein family; CpBBX19 belonged to the IV subfamily and was closely related to AtBBX19. In addition, the promoter region 2000 bp upstream of the *CpBBX19* gene was analyzed (Figure 1c and Appendix A). The *CpBBX19* promoter region contained one defense and stress responsive element (TC-rich), one element involved in salicylic acid responsiveness (TAC-element), one MYB binding site (MBS) involved in drought-inducibility, three elements related to abscisic acid responsiveness (ABRE), and four elements involved in MeJA responsiveness (TGACG-motif and CGTCA-motif), indicating that *CpBBX19* might be related to the stress response in plants.

As shown in Figure 2, the six conserved motifs in *CpBBX19* and the IV subfamily of *Arabidopsis*, identified by MEME software, comprised between 8 and 47 amino acids. Sequence analysis revealed that motif 1 contained the complete structure C-X2-C-X8-C-X7-C-X2-C-X4-H-X8-H of B-Box1. Motif 2 contained the C-X2-C-X8-C part of B-box2, while motifs 3 and 4 contained the last histidine (H) of B-box2 and a more conserved amino acid sequence. Motifs 2, 3, and 4 constituted a relatively complete conserved domain of B-Box2.

### 3.2. Tissue-Specific Expression of CpBBX19

qRT-PCR revealed that *CpBBX19* was expressed in the root, stem, leaf, and flower tissues of wintersweet, but the expression levels varied with location. Expression levels of *CpBBX19* were relatively high in cotyledons and inner petals of wintersweet, and in other floral organs (Figure 3a). The expression patterns of *CpBBX19* were also explored during the different developmental stages of wintersweet flowers (Figure 3b). The highest level of *CpBBX19* expression was detected in the early initiating bloom stage, followed by late initiating bloom and bloom, and the lowest levels of expression were present in the stages of germination, flower-bud, petal-display, and withering.

### 3.3. Expression Profiles of CpBBX19 under Abiotic Stresses and Hormone Treatments

To explore the effect of stresses and exogenous hormones on *CpBBX17* expression, qRT-PCR was used to determine the transcription level of *CpBBX19* in wintersweet under different abiotic stresses (drought, salt, cold, and heat) and hormone treatments (ABA and MeJA). These treatments were selected based on the *cis*-acting elements in the *CpBBX19* promoter region (Appendix A). For the ABA, MeJA, and NaCl treatments, expression of *CpBBX19* was induced and peaked at 2 h (Figure 4a,b,e). For heat treatment, *CpBBX19* expression increased and reached a peak at 2 h, then gradually decreased and was lower than that of the control (untreated; CK) at 24 h. (Figure 4c). For cold treatment, the expression of *CpBBX19* was significantly higher than that of the untreated control at 6 h and 12 h (Figure 4d). Under PEG-6000 treatment, expression of *CpBBX19* was induced and increased between 2 h and 6 h, then decreased from 6 h to 12 h, before increasing again and peaking at 24 h (Figure 4f).

### 3.4. Overexpression of CpBBX19 Improves Tolerance to Salt and Drought Stress in Arabidopsis

To further study the function of *CpBBX19*, the expression vector pGWB551*-CpBBX19* was constructed and then transferred into wild-type *Arabidopsis thaliana* (WT). Using the leaves DNA of the T2 lines as templates, the *CpBBX19* overexpression lines were identified by PCR (Appendix A), and WT was used as control. Three lines with high, medium, and low expression levels of *CpBBX19* (namely, OE-2, OE-5, and OE-9; Figure 5) were selected for further functional analysis. The relative expression levels of OE-2, OE-5, and OE-9 were 1.104 ± 0.080, 0.460 ± 0.019, and 0.211 ± 0.029, respectively. Wild-type plants (WT) were used as a control.

To determine the response of *CpBBX19* overexpression to NaCl stress, the three transgenic lines and WT plants were irrigated with 300 mM NaCl solution. After 4 days, WT and transgenic lines all showed obvious wilting, but the growth status of the transgenic lines was slightly better than that of WT (Figure 6a). Electrolyte leakage and malondialdehyde (MDA) are reference indicators of the degree of plant damage [48,49]. Electrolyte leakage of the three transgenic lines was 65.24%, 68.76%, and 69.35%, respectively, all significantly lower than that of WT (85.27%) (Figure 6b). In addition, MDA molar concentrations of the transgenic lines (0.409, 0.6785, and 0.701 μmol/g∙Fw, respectively) were markedly lower than that of WT (0.929 μmol/g∙Fw) (Figure 6c). Therefore, transgenic lines have better resistance to damage under salt stress compared with WT plants.

To explore the role of *CpBBX19* in drought stress, WT plants and the three transgenic lines were irrigated with 30% PEG. After 36 h, WT plants were markedly wilted compared with transgenic plants (Figure 7a). Measurement of electrolyte leakage showed that WT was the most damaged (67.37% electrolyte leakage) among the analyzed plants, and this measurement was significantly different from that of the three transgenic lines (31.81%, 48.87%, and 62.30%, respectively) (Figure 7b). MDA content in WT plants (0.678 μmol/g∙Fw) was almost three times that of OE-2 plants (0.226 μmol/g∙Fw), and twice that of OE-5 (0.360 μmol/g∙Fw) and OE-9 (0.430 μmol/g∙Fw) plants (Figure 7c). In summary, overexpression of *CpBBX19* improved the drought tolerance of transgenic *Arabidopsis*.

## 4. Discussion

Transcription factor regulation is a crucial link in gene expression. B-box-like zinc-finger proteins in animal cells are involved in many cellular processes, such as apoptosis, cell cycle regulation, virus response, and other intracellular physiological processes [50,51]. Research on the structure and function of B-box-like proteins in plants has also made some progress. BBX family members have been identified in *Arabidopsis* [16], rice [30], apple [27], *Toona Sinens* [24], wheat [52], pear [25], and *Petunia hybrida* [53], with the number of BBX members varying among different species. Wintersweet is an ornamental plant that blossoms in winter in China. However, there are no reports on *BBX* transcription factors of wintersweet. In this study, a novel *BBX* transcription factor gene, *CpBBX19*, was isolated based on information from the wintersweet transcriptome database [39]. Multiple sequence alignment revealed that CpBBX19 protein shared a high sequence similarity with the BBX19 protein sequences of other plants. Two B-box domain motifs were found in the CpBBX19 sequence (Figure 1a), which was consistent with the characteristics of group IV members of BBX proteins. Phylogenetic analysis showed that CpBBX19 clustered with group IV BBX proteins and was closely related to AtBBX19 (Figure 1b). Prediction of the subcellular localization of protein CpBBX19 indicated that it was located in the nucleus, and it was speculated that this transcription factor might be involved in the transcription level of other genes.

Gene function can be reflected by the expression patterns of a gene to a certain degree [42]. *OsBBX13* had the highest expression in leaves, then in roots, and lower expression in stems and young panicles [44]. *CmBBX19* had higher transcript contents in leaves, flowers, and stems, but lower levels in roots [54]. *PuBBX24* was expressed in the roots, stems, leaves, flowers, and fruits of *Pyrus ussuriensis*, but expression levels were highest in stems and lowest in leaves [55]. In the current study, *CpBBX19* was expressed in the stems, roots, leaves, and flowers of wintersweet, with highest expression found in cotyledons and inner petals and lower expression detected in roots, stems, and old leaves. The expression patterns of *PuBBX24, CmBBX19*, and *OsBBX13* were slightly different from that of *CpBBX19*, but all four genes exhibited higher expression in leaves and flowers and lower expression in roots. We speculated that the gene *CpBBX19* may be involved in leaf morphogenesis and may play a role in the inner petal formation of wintersweet flowers. At the stage of floral development, the highest expression of *CpBBX19* was in the early flowering stage, followed by late initiating bloom and bloom, and the lowest level of expression was detected in the stages of germination, flower-bud, petal-display, and withering. As wintersweet blossoms in cold winter and high expression of *CpBBX19* was detected in the early initiating bloom stage, this gene may be related to the stress-resistant protection mechanism of wintersweet against adverse environments. *CpBBX19* may be involved in initiating the relevant regulatory network to enhance the stress resistance of floral organs.

*Cis*-acting elements play a key role in gene transcription and expression [41]. Most of the *cis*-acting elements with biological functions are predominantly active 50 bp upstream of the transcription start site (TSS), and most of the transcription factor binding sites (TFBSs) are in the region from −1000 bp to +200 bp with respect to the TSS [56]. Analysis of the promoter of *CpBBX19* revealed the presence of a TATA-box, which is a core promoter element usually located 25–35 bp upstream of the TSS [57,58,59,60]. The promoter of *CpBBX19* also contained a CAAT-box, which also commonly exists in promoters and is usually located 75 bp upstream of TSS [57,58,59,60]. Many *cis*-acting elements of abiotic stresses and hormone responses, such as ABRE, ARE, MBS, MYB, MYC, TC-rich, TCA-element, and TGACG-motif, were also present. These observations indicated that *CpBBX19* may be involved in abiotic stress and hormone responses. Expression of the *BBX* gene family members *SlBBX7*, *SlBBX9*, and *SlBBX20* was induced by cold stress in tomato [61], and there were differences in the expression of some *GhBBX* genes after ABA treatment in *Gossypium hirsutum* [62]. *PavBBX6* and *PavBBX9* showed strong induction under ABA, GA, and BR treatment in sweet cherry fruit [63]. Furthermore, abiotic stress such as drought, low temperature, and salt stress strongly induced expression of *OsBBX1*, *OsBBX2*, *OsBBX8*, *OsBBX19*, and *OsBBX24* [31], and expression of *OsBBX2*, *OsBBX7*, *OsBBX17*, *OsBBX19*, and *OsBBX24* was upregulated by GA, SA, and MeJA hormones at different time points [31]. These results are consistent with those of *CpBBX19*, with the expression of *CpBBX19* being induced by abiotic stress (drought, salt, cold, and heat) and exogenous hormones (ABA and MeJA) (Figure 4) in the current study. Therefore, *CpBBX19* may be involved in the response of wintersweet to drought, salt, cold, heat, ABA, and MeJA. However, the expression pattern of *BBX* genes in some plants was opposite to the above pattern. For example, expression of *CmBBX19* was downregulated by drought stress and ABA treatment [54], and the expression of *GhBBX5*, *GhBBX23*, and *GhBBX28* in cotton was continuously downregulated after PEG treatment for 1 h, reaching the lowest expression point at 12 h, and then increased at 24 h [62]. These differences suggested that *BBX* genes have different expression patterns in different plants.

Electrolyte leakage and MDA content, as an important index of plant stress resistance, can reflect the degree of damage of the plant cell membrane system under adversity [64]. The increase of electrolyte leakage and MDA content indicates the cell membrane structure damage and intensified membrane permeability [65]. To verify our hypothesis and facilitate understanding of the functions of *CpBBX19*, *Arabidopsis* plants overexpressing *CpBBX19* were obtained. Growth of the transgenic lines was better than that of WT lines under drought stress, and the electrolyte leakage and MDA content of these transgenic lines were much lower than those of WT lines. Under salt stress, all plants (WT and OE lines) were obviously withering, but the electrolyte leakage and MDA content of OE lines were significantly lower than those of WT. This indicated that the transgenic *Arabidopsis* plants had enhanced salt and drought resistance compared with the WT plants. Some functions of *BBX* genes have been verified. For example, *CmBBX24* transgenic *Arabidopsis* was more tolerant to low temperature and drought stress than the WT lines were [66]; transgenic *CmBBX22* in *Arabidopsis* improved the drought resistance of *Arabidopsis* [38]; heterologous expression of *MdBBX10* enhanced the tolerance of transgenic *Arabidopsis* to salt and drought stress [28], and compared with WT plants, overexpression of *AtBBX24* could improve the tolerance of *Arabidopsis* to salt stress [67]. However, because of the inclusion of an EAR motif in *CmBBX19*, the drought tolerance of *CmBBX19*-overexpressing (*CmBBX19*-OX) lines was lower than that of WT [54]. In plants, proteins containing an EAR motif have been proven to be transcriptional repressors and are instrumental in many biological processes, including responses to hormones and biotic and abiotic stresses, floral transition, and meristem maintenance [68,69]. In general, different BBX transcription factors have complex roles in plant responses to environmental stresses.

## 5. Conclusions

To date, there are no reports on the *BBX* gene family in wintersweet. In this study, *CpBBX19* from wintersweet was isolated and its expression pattern and functional characterization were investigated. Expression of *CpBBX19* was induced by multiple abiotic stresses (drought, salt, cold, and heat), as well as by hormone treatments (ABA and MeJA). Heterologous expression of *CpBBX19* in *Arabidopsis* enhanced tolerance to drought and salt stresses. These findings indicate that *CpBBX19* plays a role in drought and salt stress responses. Moreover, this study also lays a strong foundation for further investigations into *BBX* genes of wintersweet and enriches understanding of the molecular mechanism of stress resistance in wintersweet.

## Figures and Tables

**Figure 1 genes-12-01456-f001:**
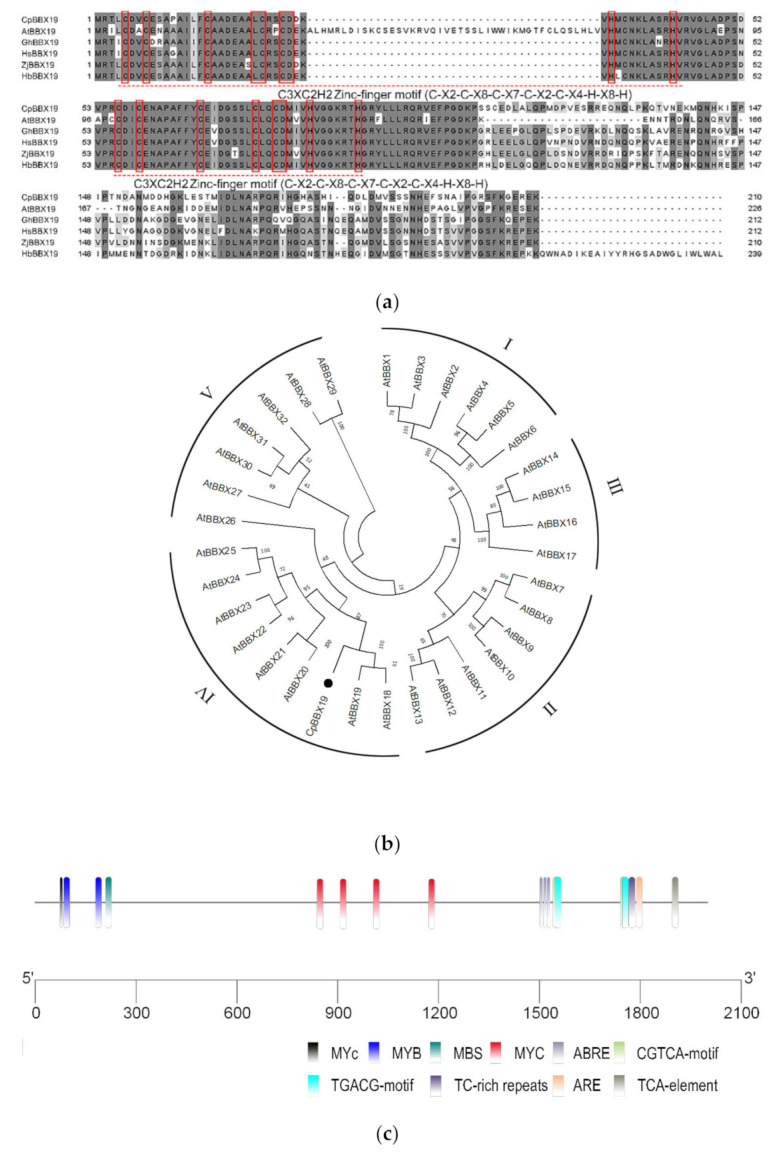
Characterization of *CpBBX19*. (**a**) Multiple sequence alignment of CpBBX19 with BBX19 proteins from *Arabidopsis thaliana* (accession number NP_001190958.1), *Gossypium hirsutum* (XP_016697629.1), *Hibiscus syriacus* (XP_039058672.1), *Ziziphus jujube* (XP_015892301.1), and *Hevea brasiliensis* (XP_021655083.1). Identical and similar amino acids are shaded in gray and light gray, respectively, and conserved zinc-finger motifs are marked by red boxes. (**b**) Phylogenetic analysis of CpBBX19 and BBX proteins from *Arabidopsis*. MEGA 6 software with the neighbor-joining (NJ) method (1000 bootstrap repeats) was used to reconstruct the phylogenetic tree. Accession numbers of the protein sequences used for phylogenetic analysis are shown in Appendix A. (**c**) *Cis*-acting elements in the promoter of *CpBBX19*.

**Figure 2 genes-12-01456-f002:**
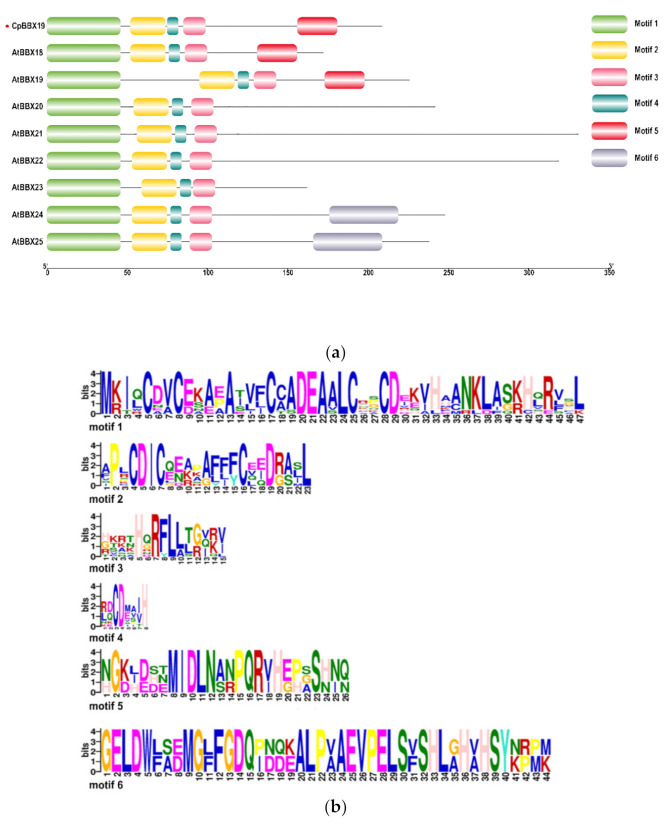
Motif distribution in CpBBX19 and the IV subfamily of the BBX family of *Arabidopsis*. (**a**) Motif locations. (**b**) The width and amino acid sequence of conservative motifs. CpBBX19 is marked by a red point.

**Figure 3 genes-12-01456-f003:**
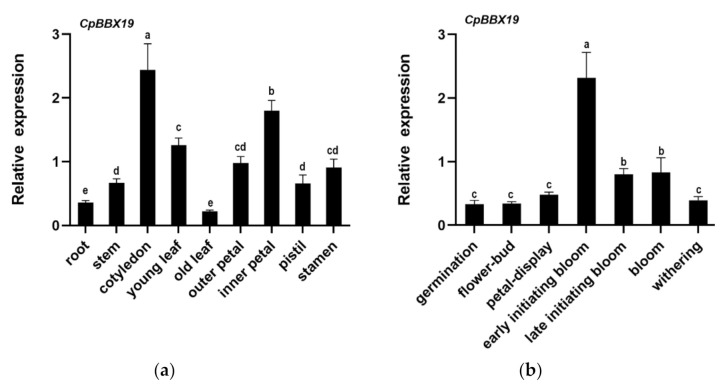
Expression pattern analysis of *CpBBX19* in wintersweet by qRT-PCR. (**a**) Expression of *CpBBX19* in different tissues of wintersweet. (**b**) Expression of *CpBBX19* in different floral development stages of wintersweet. *CpActin* was used as the internal reference gene. Data represent the mean of three biological repeats ± SD. Error bars indicate standard deviation. Different lowercase letters (a–e, cd) on bars indicate significant differences (*p* < 0.05).

**Figure 4 genes-12-01456-f004:**
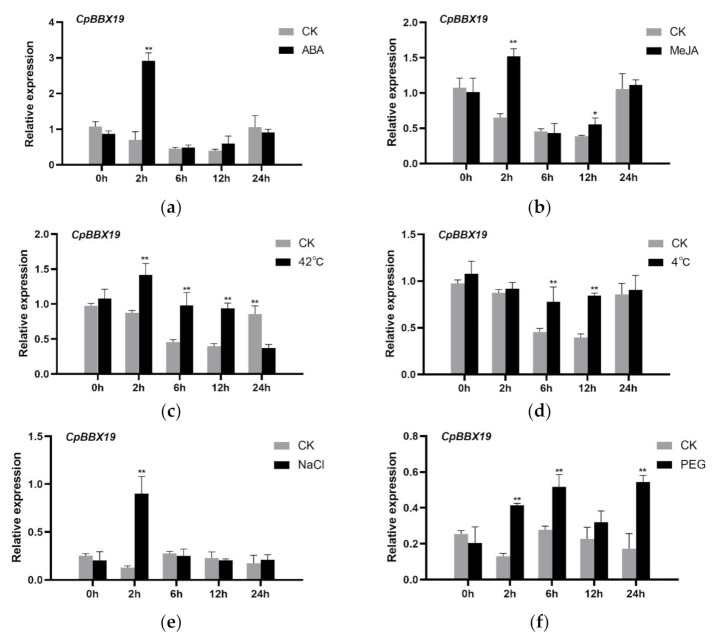
Expression profiles of *CpBBX19* in response to abiotic stress and hormone treatments. Six-leaf stage wintersweet was exposed to (**a**) 50 µM ABA, (**b**) 100 µM MeJA, (**c**) 42 °C, (**d**) 4 °C, (**e**) 300 mM NaCl, and (**f**) 50% PEG-6000 treatments. RNA was extracted from the leaves at 0, 2, 6, 12, and 24 h after treatment. *CpActin* was used as the internal reference gene. Data represent the mean of three biological repeats ± SD. Error bars indicate standard deviation. **p* < 0.05, ** *p* < 0.01.

**Figure 5 genes-12-01456-f005:**
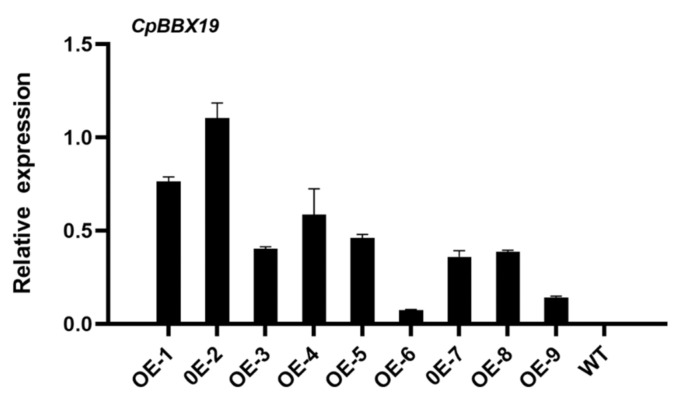
qRT-PCR identification of *CpBBX19* transcript levels in the leaves of nine overexpressing *Arabidopsis* lines and WT plants. *AtActin* was used as the internal reference gene. Bars indicate the SE of the mean from three technical replicates and three biological replicates.

**Figure 6 genes-12-01456-f006:**
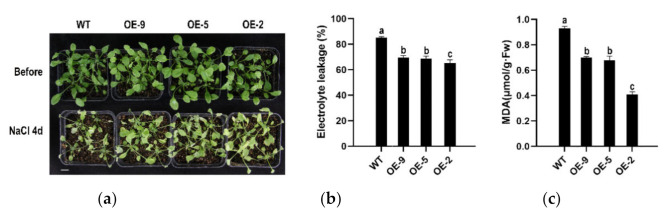
Heterologous expression of *CpBBX19* enhanced salt tolerance of transgenic *Arabidopsis* plants. (**a**) Performance of *Arabidopsis* overexpressing *CpBBX19* (OE lines) under NaCl treatment. Bar denotes 1 cm. (**b**) Electrolyte leakage of WT and OE lines under NaCl treatment. (**c**) MDA content of WT and OE lines under NaCl treatment. Data represent the mean of three biological repeats ± SD. Error bars indicate standard deviation. Different lowercase letters (a, b, c) on bars indicate significant differences (*p* < 0.05).

**Figure 7 genes-12-01456-f007:**
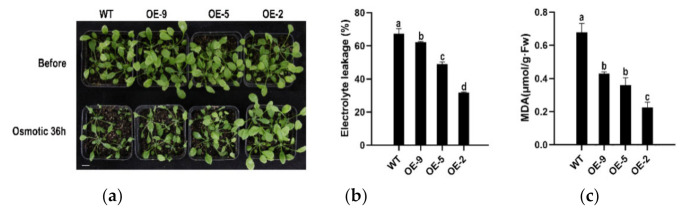
Heterologous expression of *CpBBX19* enhanced drought tolerance of transgenic *Arabidopsis*. (**a**) Performance of *Arabidopsis* overexpressing *CpBBX19* (OE lines) under 30% PEG-6000 treatment. Bar denotes 1 cm. (**b**) Electrolyte leakage of WT and OE lines under 30% PEG-6000 treatment. (**c**) MDA content of WT and OE lines under 30% PEG-6000 treatment. Data represent the mean of three biological repeats ± SD. Error bars indicate standard deviation. Different lowercase letters (a, b, c) on bars indicate significant differences (*p* < 0.05).

## Data Availability

Not applicable.

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
