# Peer review of "CpBBX19, a B-Box Transcription Factor Gene of Chimonanthus praecox, Improves Salt and Drought Tolerance in Arabidopsis"

_genes, 2021, doi:10.3390/genes12091456_

Round 1

Reviewer 1 Report

Comments for the Authors

In the manuscript ID: genes-1366083CpBBX19, a B-box Transcription factor gene of Chimonanthus praecox, improves salt and drought tolerance in Arabidopsis“, the authors Wu et al. reported identification and characterization of the transcription factor CpBBX19 in Chimonanthus praecox. The authors concluded that the transcription factor CpBBX19 is involved in plant tolerance of Arabidopsis plants to abiotic stress.

The manuscript contains new and valuable information, and the data can serve as a basis for further understanding of molecular mechanisms of plant tolerance to abiotic stress. However, the text and figures suffer from shortcomings and/or other errors that must be remedied. The manuscript is carelessly written, it abounds with some typing errors and some incomprehensible sentences and very fundamental flaws. English language needs to be improved.  

Materials and Methods

Most of the paragraphs of the chapter lack detailed description of the methods. For example, in the chapter 4.1, there is written that the experimental plants grew “under natural conditions in campus of … .” It should be mentioned at least in what time of year the material was collected. There is information, that mature plants were used for the experiments. I am wondering if mature plants of wintersweet contain cotyledons, which were used for experiments? How experimental plants were old? In this manuscript, the expression of CpBBX19 gene was studied as a function of various abiotic stress and hormone treatment. However, the complete and detailed description how plants were exposed to the stress conditions and hormones (heat, NaCl, cold, PEG, ABA, MeJA) is missing in the chapter Material and Methods. Some very brief information are mentioned in the figure legends but it is not sufficient in any case.  

Legend of Table S1 is not sufficient.

References

The references in the list of references are mot written in uniform format. In many references, the page number or article number is missing. In the reference no. 9, author names are not mentioned, initials only. In the reference no. 10, the journal name is written in capital, etc.

Minor errors

Table S2 does not show primer sequences as mentioned in the paragraph 4.4. It should be Table S1 instead.

The abbreviation MDA is not explained in the manuscript.

In the chapter Results, Figure S1 and Figure 2 are mentioned. These figures are not part of this manuscript. There are Tables S1 and S2 available.

Reviewer 2 Report

Results presented by Authors provide novel informations related to salt/drought stress response in A. thaliana overexpressing the CpBBX19 gene.

Results are significant, however the article needs following corrections:

Introduction section:

Authors may add several sentences describing typical plant  response to salt/drought stress in context of electrolyte leakage and MDA concentration.  

Line 102:

The MZ740468 record can not be found in the NCBI database- please provide the correct accession number or provide name of other database where this sequence is deposited.

In the sentence is used twice the word “database”- correct it.

Lines 101-107: If the ORF is 633bp then the full length cDNA should be the same. Putatively the 1563bp sequence contains also 5’ promoter fragment or 3’ regulatory elements. Authors should state if the 1536bp fragment contains also gene  regulatory 5’ or 3’ sequences.

Line 116- how the 2000bp-long promoter sequence was obtained? Describe it in material and method section.

Lines 116-122

The concept of finding cis-active motif in promoter region to support the response of promoter to particular stimuli is generally ok. However, Authors should take under consideration (write one-two sentences about it in discussion or result section) that most of biologically functional cis-active elements are within the proximal promoter (or 5’UTR)- about 250bp starting  from transcription initiation site. For example see following article:

Yu ChP, Lin JJ, Li WH (2016) Positional distribution of transcription factor binding sites in Arabidopsis thaliana. Sci Rep 6:25164.

Fig1c- try to use more contrast colors to mark particular cis-active elements.

Lines 194-195:

Authors should present results of RT-PCR experiments showing quantitative difference in CPBBX19 gene expression between A. thaliana transgenic lines. The sentences high, medium or low expression are not precise.

Section 4.2- Gene isolation; provide details of PCR cycling.

More precise description of RT-PCR experiments is necessary, following points could be added:

Assessment of RNA purity and concentration

Details of DNase treatment to remove putative remnants of genomic DNA

Reverse transcription reaction; details of reaction-temperature and time, volume and amount of used RNA

qPCR target- length of PCR product (control and tested gene), target gene symbol and accession number,

qPCR protocol:  concentration of magnesium ions, dNTPs, DNA polymerase type and concentration, additives as SYBR Green, DMSO etc. Name and manufacturer of qPCR instrument.

Reason to choose the RT-PCR control gene (actin) as internal standard- citation of previous research or analysis using bestKeeper tool (Pfaffl et a. 2004)

Section 4.5

Some information of the promoter in the overexpression vector pGWB551 should be provided as this construct mediate of the overexpression of studied CpBBX19 gene in A. thaliana. Is this constitutive CaMV 35S or related constitutively active promoter? Authors may also provide a simple map of this vector as an supplementary file.

Section 4.6

Authors should present the plant transformation by independent method, that may be performed relatively quickly, for example by isolating genomic DNA from transformed and untransformed plants (available kits for plant genomic DNA isolation or using method of Khan et al. 2007 ) and applying primers specific to CpBBX19 gene that should produce specific PCR product that may be visible as DNA band of expected size only in transformed plants.

Khan S, Qureshi MI, Kamaluddin AT, Abdin MZ (2007) Protocol for isolation of genomic DNA from fresh and dry roots of medicinal plants suitable for RAPD and restriction digestion. Afr J  Biotechnol 6:175-178.

Table S2- Authors write of 55 TATA box and 25 CAAT motifs. The TATA box, if exist (majority of plants promoter does not contain it) is localized about 325-35bp 5’ from transcription start site. Also the CAAT motif is localized 60-100 bp 5’ upstream from transcription start site. So the TATA-box and CAAT-box (if exist) is generally one per promoter. Compare following reference:

Molina C, Grotewold E (2005) Genome wide analysis of Arabidopsis core promoters. BMC Genomics 6:25. doi: 10.1186/1471-2164-6-25

Line 374; sentence should be for example as follows: “MDA was determined as described by Robert [38].”

Round 2

Reviewer 1 Report

The authors added all the necessary information based on the opponent's comments.

Author Response

Thank you very much for your careful work and guidance.

Reviewer 2 Report

Thank you for corrections that significantly improved the manuscript. I have found following  minor mistakes, mainly of typographic nature, that should be corrected:

In lines 214-217 Authors describe the confirmation of transgenic material in genomic DNA by PCR. Authors may add here details of PCR cycling reaction. However in lines 328-330 Authors wrote that confirmed by RT-PCR. In results section Authors should also write one-two sentences of confirmation by PCR described in materials and methods. If Authors have a good quality gel picture confirming the plant transformation (described as above in material and methods), it could be shown as a supplement file.

Line 137: is nM; should be nm

Line 138: is uM, should be μM. Make this correction also in other places.

Line 138: is uL; should be μL. Make this correction also in other places.

Line 155; place between two brackets is double spaced- make only one.

Line 161- is μl should be μL

Line 179; check out if instead of 300 μM should not be 300 mM NaCl. The 300 μM NaCl is too low concentration to make any salt/drought stress. The same for line 324. Compare line 338- there is 300 mM as it should be.

Line 192- it should be MgCl2 - not MgCl2

Line 200- provide in text the length of reference gene (actin)- 200bp.

Line 332; Authors may provide in text (one-two sentences) the values - mean and standard deviation of relative expression for three lines-high, medium and low expression of CpBBX19

Line 415; I mistakenly provided you the wrong number 325-35. It should be 25-35. Correct it.

Line 417 ; instead of 60-100 it should be 75. It will be more precise.

To better support above corrections in lines 415 and 417 include please two more citations (basic data below to find them) - add them to citations nr 58 and 59 in lines 415 and 417.

Florquin K. …..  Nucleic Acids Research 2005, 33:4255-4264 (Large-scale structural analysis of the core promoter  in mammalian and plant genomes)

Shahmuradov I…..Nucleic Acids Research 2003, 31:114-117 (PlantProm: a database of plant promoter sequences)

Some corrections in references section:

Reference 29: names of Authors should be: Kiełbowicz-Matuk, A.; Rey, P.; Rorat, T (check-out in article and correct it). Authors wrongly wrote Agnieszka and then only initials of Author names.

Reference 31 and 34 are the same, remove one of them. Instead of Bmc Genomics should be BMC Genomics. After the nr of volume (20) should be the electronic number of article- here it should be 27.

Reference 41- lacking nr of volume (only year and nr of pages is provided).

Reference 56- capitalized entire names of Authors

Reference 58 - Instead of Bmc Genomics should be BMC Genomics

Reference 59; lacking volume nr

Reference 60- lacking electronic nr of article

References 63 and 64- capitalized entire names of Authors
